# Recursive Autoregressive Depth Estimation with Continuous Token Modeling

## Abstract

Monocular depth estimation is a cornerstone of robotic perception and computer vision, yet reconstructing 3-D structure from a single RGB image suffers from severe geometric ambiguity and uncertainty. Motivated by the recent success of autoregressive (AR) models in image generation, we introduce a Fractal Visual AR + Diffusion framework that predicts depth both accurately and efficiently. Conventional pixel-wise AR generation is too slow for robotic applications, so we design a coarse-to-fine, multi-scale autoregressive pipeline: the model first sketches a global depth map at low resolution and then refines it progressively to full pixel fidelity, greatly accelerating inference. To bridge the RGB–Depth modality gap, each scale incorporates a Visual-Conditioned Feature Refinement (VCFR) module that fuses multi-scale image features with the current depth prediction, explicitly injecting geometric and textural cues. Because discretizing continuous depth values can cause information loss and unstable training, we adopt a conditional denoising diffusion loss that models depth distributions directly in continuous latent space, fundamentally avoiding quantization errors. Although the visual AR–diffusion paradigm boosts accuracy, its layer-by-layer generation still introduces latency. To reclaim speed, we abstract the Visual AR unit into a reusable base generator and invoke it recursively, forming a self-similar fractal architecture that preserves modeling power while cutting the inference path.

## 1 Introduction

Monocular depth estimation is a fundamental task in robotics perception and computer vision, playing a critical role in scene understanding and reconstruction Izadi et al. (2011); Chen et al. (2019). It is widely applied in domains such as autonomous driving Wang et al. (2019). The objective is to predict per-pixel depth values from a single RGB image. Most deep learning based approaches adopt a top-down and bottom-up encoder–decoder architecture Eigen et al. (2014); Agarwal & Arora (2023) to perform depth estimation.

Although autoregressive (AR) generative models have demonstrated impressive performance in image generation tasks, directly applying them to monocular depth estimation poses several significant challenges. First, there exists a clear mismatch between the input and output modalities: unlike traditional image generation, which typically involves mapping from one image to another image within the same modality, monocular depth estimation requires predicting a geometric depth map from an RGB image. This modality discrepancy makes it difficult for the model to effectively capture cross-modal correlations. Second, the pixel-by-pixel generation process in traditional AR models results in low inference efficiency. In practical applications such as robotic perception, where real-time performance is crucial, slow inference severely limits the deployment of such models. To address these issues, we propose a novel framework based on a "next-scale prediction" strategy, which reformulates monocular depth estimation as a progressive autoregressive generation process from low to high resolution. This strategy significantly improves the generation

efficiency by predicting depth hierarchically across multiple spatial scales. To further alleviate the challenges brought by modality mismatch, we introduce the Visual-Conditioned Feature Refinement (VCFR) module. VCFR extracts multi-scale visual features from the input image and fuses them with the current scale's depth predictions to form a Visual-Depth Joint token, which serves as a conditional input for predicting depth at the next scale. This design greatly enhances the model's ability to capture geometric structures and fine-grained texture details across modalities. In addition, another major challenge of traditional AR models lies in the quantization process. Converting continuous depth values into discrete tokens often results in information loss and unstable training. However, the core principle of autoregression, predicting the next token conditioned on the previous one, is not inherently tied to whether the values are discrete or continuous Li et al. (2024a). Therefore, we incorporate a conditional denoising diffusion loss, leveraging the strong distribution modeling capability of diffusion processes in continuous space. Conditioned on the Visual-Depth Joint token, our model directly models the depth distribution at each scale, effectively mitigating quantization errors caused by discrete tokenization. Nevertheless, despite the improved accuracy brought by combining visual autoregression with diffusion modeling, the iterative multi-scale generation process still introduces inference latency. To further enhance efficiency, we adopt a fractal architecture, where the model is recursively composed of smaller-scale autoregressive modules. This design maintains modeling capacity while significantly accelerating inference.

We summarize our contributions as follows: 1. We propose a novel fractal-based depth estimation framework. We construct an autoregressive monocular depth estimation model in which each autoregressive unit is recursively composed of smaller autoregressive modules. 2. We introduce the Visual-Conditioned Feature Refinement (VCFR) module. During each scale's autoregressive prediction, VCFR aggregates visual features from the image and combines them with current scale depth predictions to generate a Visual-Depth Joint token. 3. We incorporate a conditional denoising diffusion loss into the scale-wise autoregressive process, replacing traditional vector quantization. By conditioning on the Visual-Depth Joint token, our model directly learns the depth distribution at the next scale in continuous space through the diffusion process. Our framework is illustrated in Fig. 1.

## 2 RELATED WORK

### 2.1 MONOCULAR DEPTH ESTIMATION

**Supervised Learning:** Supervised learning approaches treat depth estimation as a per-pixel regression task, trained on datasets with ground-truth depth annotations. Early studies were mostly CNN-based, such as RAP Zhang et al. (2019) and DAV Huynh et al. (2020), which improved depth prediction by refining feature extraction modules. Later, Transformer-based architectures were introduced, e.g., PixelFormer Agarwal & Arora (2023). Zhang et al. (2025a) designs an image-fusion method better tailored for depth estimation, and Zhang et al. (2025b) performs depth estimation based on defocus cues. Another line of work formulates depth estimation as a regression–classification hybrid to model depth distributions more precisely. Methods such as AdaBins Bhat et al. (2021) adopt adaptive binning strategies that combine classification and regression for depth prediction. WordDepth Zeng et al. (2024) is built on a VAE framework, using textual descriptions to predict a depth distribution; samples drawn from images are then combined with this distribution to produce depth. Building on this idea, Zhang & Lu (2025) further exploits a camera model to compute depth priors from environmental information, from which depth-related textual descriptions and depth-aware image features are extracted. Autoregressive depth estimation methods Wang et al. (2025); Gabdullin et al. (2024) based on discrete latent spaces struggle to capture the inherently continuous nature of depth, making quantization error unavoidable. To address this, we combine autoregression with conditional diffusion, directly modeling depth in a continuous latent space. We inject image features as conditioning signals into the diffusion process, yielding a tighter coupling between depth estimation and autoregressive generation.

**Self-Supervised Learning:** Self-supervised learning methods leverage the structural information of images for depth estimation without requiring ground truth depth labels. These approaches often rely on stereo matching, temporal constraints, or auxiliary self-supervised tasks. Zhang et al. (2024) uses a camera model

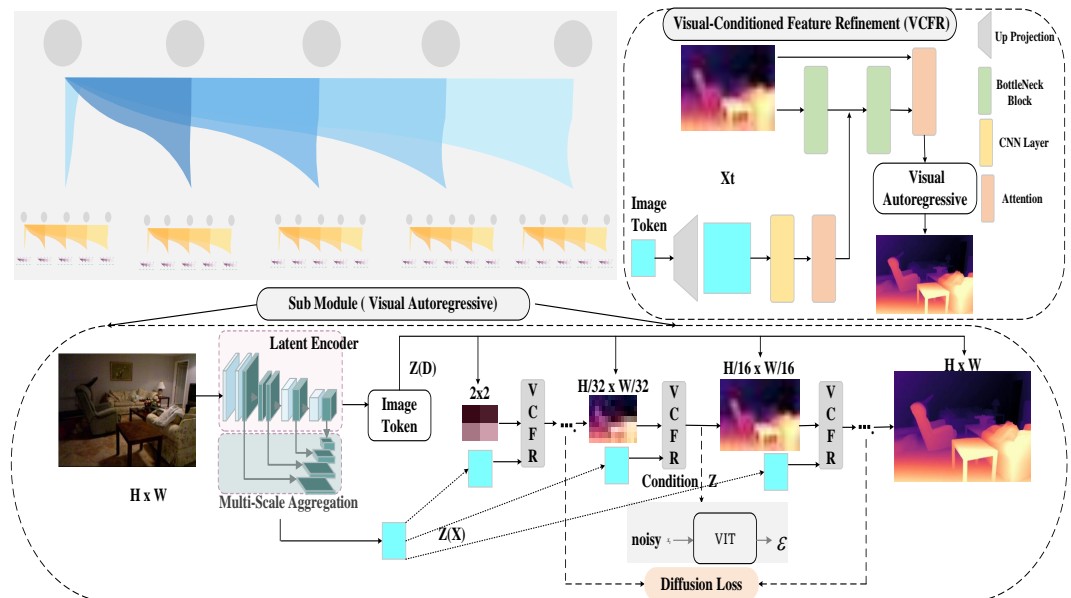

Figure 1: Overall architecture of the proposed Fractal Visual Autoregressive Diffusion model. Given an input RGB image with resolution $H \times W$, we first extract its visual features $Z(X)$ through an image encoder and multi-scale aggregation module. These features serve as global conditional guidance throughout the entire depth generation process. Our framework adopts a fractal-style recursive structure, where the generation task is divided into four nested modules $\{g_4, g_3, g_2, g_1\}$. Starting from a coarse latent token, the model progressively recovers higher-resolution depth representations. The lowest module $g_4$ takes a $1 \times 1$ latent as input and predicts the initial coarse depth latent, which is then passed as conditional input to $g_3$, and so on, until the final module $g_1$ outputs the high-resolution depth map $\hat{D} \in \mathbb{R}^{H \times W}$. In each module, instead of being generated in a single step, the depth map is predicted token-by-token via a visual autoregressive diffusion process. Each visual autoregressive diffusion module takes the current image as input token along with the depth latent token produced by the previous layer, and aims to model the continuous depth distribution at the current resolution (e.g., $H/32 \times W/32$, $H/16 \times W/16$, or the final $H \times W$). During this process, we introduce a conditional denoising diffusion mechanism. The noise predictor $\epsilon_\theta(x_t \mid t, z)$ receives the current diffusion timestep $t$ and condition vector $z$, and predicts the noise perturbation, which is used to optimize the per-token continuous depth distribution via Diffusion Loss. The top-left part shows our fractal framework.

to compute depth priors to assist self-supervised depth estimation. ZoeDepth Bhat et al. (2023) and Depth Anything Yang et al. (2024a) further adopted self-supervised strategies, significantly enhancing model generalization. In addition, recent research has explored incorporating multimodal information to boost depth estimation performance. For instance, VPD and related studies Patni et al. (2024); Chatterjee et al. (2024) utilize vision-language pretrained models such as CLIP to guide depth estimation.

## 2.2 GENERATIVE MODELS IN DEPTH ESTIMATION

**Diffusion Models:** Diffusion models have achieved remarkable success in image generation tasks, including unconditional image synthesis Dhariwal & Nichol (2021), text-to-image generation Ramesh et al. (2022), and image enhancement Saharia et al. (2022). However, their application in computer vision tasks such as depth estimation remains relatively limited. Recently, some studies have attempted to apply diffusion models to depth estimation. For instance, certain approaches use the denoising process of diffusion

models to refine depth predictions from input images. DiffusionDet Chen et al. (2023), for example, uses diffusion to generate detection boxes in object detection tasks. **Autoregressive Models:** Another category of generative models leverages autoregressive approaches for depth estimation. Inspired by the VQ-VAE framework, these methods typically discretize the input image and predict depth via a decoder. DORN Fu et al. (2018) was the first to propose the ordinal regression network, which discretizes the depth space into bins and performs prediction accordingly. Subsequent works, such as Ord2Seq Wang et al. (2023), modeled ordinal regression as a sequence generation task using autoregressive networks to progressively refine predictions. However, these methods still rely on the VQ-VAE paradigm, which introduces information loss due to the discretization of the input image prior to depth decoding.

## 3 METHOD

### 3.1 FRACTAL AUTOREGRESSIVE FRAMEWORK

The core idea behind the fractal generative model is to recursively construct more complex generators from existing atomic generative modules. Formally, a fractal generator $g_i$ defines how to generate a set of new outputs $\{x_{i+1}\}$ for the next-level generator based on the output $x_i$ from the previous level: $\{x_{i+1}\} = g_i(x_i)$. Since each generator can produce multiple outputs from a single input, the fractal framework enables exponential growth in generated outputs while requiring only a linear number of recursive levels. We apply the fractal generation framework to the task of depth estimation by recursively refining depth predictions from coarse to fine scales. Specifically, each recursive depth estimation module extracts conditional information from the output of the previous level and further estimates finer-grained depth structures. This fractal structure can be expressed as a recursive function call: $\{D_{i+1}\} = g_i(D_i)$, where $D_i$ denotes the depth features predicted at the $i$-th level, and $g_i$ is the depth estimation module at that level.

For example, as illustrated in Fig. 1, a depth estimation module may first predict a coarse global structure of the image, and then recursively refine each local region through similar depth estimation modules operating at finer scales. Each recursive layer is capable of producing multiple refined outputs from a single input, resulting in a hierarchical depth estimation framework with self-similar structure. Supposing we aim to model the joint distribution of all pixel-wise depth values $d_1, \cdots, d_N$ in the image, using a single autoregressive model to process such a large number of variables is computationally expensive. To address this, we adopt a divide-and-conquer strategy by abstracting the autoregressive depth estimator as a modular unit that models a conditional distribution $p(x \mid c)$. We partition the image space into several local regions of fixed size $k$, and assume the total number of pixels $N = k^n$, where $n = \log_k(N)$ is the number of recursive levels. The original depth estimation task is gradually decomposed into smaller sub-tasks, each handled by a depth estimation module at the corresponding recursion level, forming a hierarchical recursive structure. The full-image depth prediction is factorized as:

$$p(d_1, \cdots, d_{k^n}) = \prod_{i=1}^{k} p\left(d_{(i-1)\cdot k^{n-1}+1}, \cdots, d_{i\cdot k^{n-1}} \mid d_{\text{coarse}}\right), \tag{1}$$

where $d_{\text{coarse}}$ represents the coarse-scale depth information provided by the previous-level generator. Each local region is further recursively subdivided into smaller scales until fine-grained depth predictions at the pixel level are obtained. By recursively applying this divide-and-conquer process, our fractal depth estimation framework enables efficient modeling of depth information using multiple levels of submodules. Each level only needs to handle a manageable portion of the overall task, significantly reducing computational overhead compared to a monolithic autoregressive model.The recursive structure explicitly captures the inherent hierarchical nature of depth, allowing for more natural and efficient modeling of scene geometry.

### 3.2 DEPTH ESTIMATION WITH VISUAL AUTOREGRESSIVE MODELING

In autoregressive models, the encoder first generates an image feature map token, where each feature vector $f(i, j)$ has bidirectional dependencies, reflecting strong spatial correlations. However, such bidirectional

correlation conflicts with the unidirectional dependency assumption in traditional autoregressive models, where each token $x_t$ is assumed to depend only on its prefix $(x_1, x_2, \ldots, x_{t-1})$. Furthermore, the flattening operation disrupts the inherent local spatial structure of the feature map Tian et al. (2024). To address these challenges in autoregressive tasks, when applying autoregressive models for depth estimation, we follow the model in Tian et al. (2024), shifting the model objective from "predicting the next token" to "predicting the next scale". Specifically, the autoregressive unit is no longer a single pixel but a complete token map. We first generate a token map $f \in \mathbb{R}^{h \times w \times C}$ into $K$ multi-scale token maps $(r_1, r_2, \ldots, r_K)$, where each map has progressively higher resolution $(h_k \times w_k)$, and the final scale $r_K$ matches the original feature map resolution $(h \times w)$. In this setting, the autoregressive likelihood can be written as:

$$P(r_1, r_2, \ldots, r_K) = \prod_{k=1}^{K} P(r_k \mid r_1, r_2, \ldots, r_{k-1}) \tag{2}$$

where, each autoregressive unit $r_k \in [V]^{h_k \times w_k}$ represents the token map at scale $k$, containing $h_k \times w_k$ tokens, while the sequence $(r_1, r_2, \ldots, r_{k-1})$ serves as the "prefix" for $r_k$. During the $k$-th autoregressive step, all $h_k \times w_k$ tokens are generated in parallel, conditioned on the prefix sequence and the positional embedding map at scale $k$. By adopting a progressive scale-wise prediction strategy, the model ensures that each scale $r_k$ depends only on its preceding sequence $r_{<k}$, satisfying the mathematical prerequisites of autoregressive modeling and enabling a coarse-to-fine optimization of feature representation. Furthermore, this structure eliminates the need for flattening operations, thus preserving the spatial correlation among all tokens within each token map $r_k$.

### 3.3 Visual-Conditioned Feature Refinement

As described in Section 3.2, we formulate depth estimation as a scale-wise autoregressive process, aiming to predict depth information from input images rather than reconstructing the images themselves. Although autoregressive models inherently capture local pixel dependencies, relying solely on depth features from the preceding scale may fail to directly leverage rich visual structures and fine details from the input images, potentially resulting in missing geometric structures and texture details in the estimated depth. Therefore, to compensate for these deficiencies, we introduce a visual guidance module at each scale of the autoregressive prediction. This module takes aggregated visual features $c$ and the depth predictions from the current scale as inputs to the model and generates depth features at the subsequent scale. Specifically, we employ a Swin Transformer Liu et al. (2021) backbone to extract multi-scale visual features from the input images, effectively capturing both coarse structural layouts and fine-grained details of the scenes. We then enhance the interactions and integration between multi-scale features through a hierarchical aggregation and heterogeneous interaction mechanism (HAHI Li et al. (2023)). Subsequently, a feature pyramid network Lin et al. (2017) aggregates these multi-scale features into a unified monocular visual condition. Considering inference efficiency, we design a lightweight denoising head. The low-resolution visual condition $c$ exhibits significant local spatial correlations with the depth features $x_t$ to be estimated. Therefore, we first employ a local projection layer to upsample $c$ and match the spatial dimensions of the autoregressive multi-scale depth representation, while preserving local structural consistency. Subsequently, the projected conditional features undergo feature modeling through a convolutional module combined with a self-attention layer and are element-wise fused with the depth features $x_t$. Finally, the fused features are further processed by a bottleneck structure (BottleNeck He et al. (2016)) and a channel-wise attention mechanism with residual connections, yielding the final depth estimation results for the current scale.

### 3.4 Diffusion Loss for Continuous Token Modeling

Li et al. Li et al. (2024a) pointed out that vector quantization is not an essential component for autoregressive models; instead, the critical aspect lies in effectively modeling the data distribution. In depth estimation tasks, depth values inherently represent continuous spatial geometric information. Employing discrete modeling inevitably introduces quantization errors, thus impairing the accuracy and continuity of the predicted

| Method | Architecture | AbsRel ↓ | Sq Rel↓ | RMSE↓ | RMSE log↓ | $\delta < 1.25$ ↑ | $\delta < 1.25^2$ ↑ | $\delta < 1.25^3$ ↑ |
|---|---|---|---|---|---|---|---|---|
| AdaBins Bhat et al. (2021) | E-B5+mini-ViT | 0.067 | 0.190 | 2.960 | 0.088 | 0.949 | 0.992 | 0.998 |
| DPT Ranftl et al. (2021) | VIT-L | 0.060 | - | 2.573 | 0.092 | 0.959 | 0.995 | 0.996 |
| P3Depth Patil et al. (2022) | ResNet-101 | 0.071 | 0.270 | 2.842 | 0.103 | 0.953 | 0.993 | 0.998 |
| NeWCRFs Yuan et al. (2022) | Swin-Large | 0.052 | 0.155 | 2.129 | 0.079 | 0.974 | 0.997 | 0.999 |
| BinsFormer Li et al. (2024b) | Swin-Large | 0.052 | 0.151 | 2.098 | 0.079 | 0.974 | 0.997 | 0.999 |
| PixelFormer Agarwal & Arora (2023) | Swin-Large | 0.051 | 0.149 | 2.081 | 0.077 | 0.976 | 0.997 | 0.999 |
| VA-DepthNet Liu et al. (2023) | Swin-Large | 0.050 | 0.148 | 2.093 | 0.076 | 0.977 | 0.997 | 0.999 |
| IEBins Shao et al. (2023b) | Swin-Large | 0.050 | 0.142 | 2.011 | - | 0.978 | 0.998 | 0.999 |
| iDisc Piccinelli et al. (2023) | Swin-Large | 0.050 | 0.145 | 2.067 | 0.077 | 0.977 | 0.997 | 0.999 |
| DCDepth Wang et al. (2024b) | Swin-Large | 0.051 | 0.145 | 2.044 | 0.076 | 0.977 | 0.997 | 0.999 |
| WorDepth Zeng et al. (2024) | Swin-Large | 0.049 | - | 2.039 | 0.074 | 0.979 | 0.998 | 0.999 |
| EcoDepth Patni et al. (2024) | ViT-L | 0.048 | 0.139 | 2.039 | 0.074 | 0.979 | 0.998 | 1.000 |
| ZoeDepth† Bhat et al. (2023) | ViT-L | 0.054 | 0.189 | 2.440 | 0.083 | 0.977 | 0.996 | 0.999 |
| DepthAnything† Zhao (2024) | ViT-L | 0.046 | - | 1.896 | 0.069 | 0.982 | 0.998 | 1.000 |
| DAR-Base Wang et al. (2024a) | ViT-L | 0.046 | 0.114 | 1.823 | 0.069 | 0.985 | 0.999 | 1.000 |
| DiffusiongDepth Duan et al. (2024) | Diffusion | 0.050 | 0.141 | 2.016 | 0.074 | 0.977 | 0.998 | 0.999 |
| Repurposing Diffusion Ke et al. (2024) | Diffusion | 0.105 | - | - | - | 0.904 | - | - |
| DepthFM Diffusion Gui et al. (2025) | Diffusion | 0.091 | - | - | - | 0.92 | - | - |
| Ours | ViT-L + Diffusion | 0.044 | 0.132 | 1.712 | 0.069 | 0.980 | 0.997 | 0.999 |

Table 1: A quantitative depth comparison using the Eigen split of the KITTI dataset Geiger et al. (2013).

depth. By contrast, directly modeling depth in continuous space enables a more natural representation of smooth depth variations, thereby yielding higher-quality depth estimations.

In our proposed scale-wise visual autoregressive framework, the depth prediction at each scale is treated as a conditional generation task over continuous values. Traditional autoregressive methods often rely on discrete token representations, which inevitably introduce quantization errors and limit the model's ability to capture fine-grained variations in depth. To address this, we incorporate the conditional denoising diffusion loss Ho et al. (2020), leveraging its powerful capacity for modeling continuous distributions to supervise the depth prediction process at each scale. Specifically, let $x^i \in \mathbb{R}^d$ denote the ground-truth depth vector at scale $i$, and let $z^i \in \mathbb{R}^D$ be the corresponding condition vector produced by the visual autoregressive network as: $z^i = f(x^1, \ldots, x^{i-1}, V)$, where $x^1, \ldots, x^{i-1}$ are the predicted depth maps from previous scales, and $V$ represents the visual features extracted from the input image. Our goal is to model the conditional distribution $p(x^i|z^i)$ at each scale. Following the denoising diffusion framework, the loss is defined as:

$$\mathcal{L}(z^i, x^i) = \mathbb{E}_{\epsilon, t}\left[\|\epsilon - \epsilon_\theta(x_t^i \mid t, z^i)\|^2\right] \quad (3)$$

where, $\epsilon \sim \mathcal{N}(0, I)$ is a Gaussian noise vector, and $t$ denotes a random time step in the noise schedule. The perturbed input $x_t^i$ is constructed by: $x_t^i = \sqrt{\bar{\alpha}_t}\, x^i + \sqrt{1 - \bar{\alpha}_t}\, \epsilon$, where $\bar{\alpha}_t$ is a predefined noise schedule. The noise estimator $\epsilon_\theta$ is a conditional network (Transformer block) that takes the perturbed input $x_t^i$ and is conditioned on both the diffusion time step $t$ and the scale-wise conditional vector $z^i$. In our visual autoregressive framework, $\epsilon_\theta$ also incorporates visual features and multi-scale depth priors from the autoregressive backbone to guide the denoising process at each scale. Unlike traditional loss functions that rely on discrete token matching, this conditional diffusion loss allows for modeling the depth distribution directly in the continuous space, enabling more accurate and smoother depth reconstruction across scales. During training, we randomly sample multiple time steps $t$ for each image to improve the utilization of the loss without recomputing $z^i$.

**Sampler.** At inference time, the depth $x^i$ is sampled from $p(x^i|z^i)$ via the reverse diffusion process:

$$x_{t-1}^i = \frac{1}{\sqrt{\alpha_t}}\left(x_t^i - \frac{1 - \alpha_t}{\sqrt{1 - \bar{\alpha}_t}}\, \epsilon_\theta(x_t^i|t, z^i)\right) + \sigma_t \delta \quad (4)$$

where $\delta \sim \mathcal{N}(0, I)$, and $\sigma_t$ is the noise level at step $t$. The process starts from $x_T^i \sim \mathcal{N}(0, I)$ and iteratively generates the final depth prediction $x_0^i \sim p(x^i|z^i)$. We also introduce a temperature parameter $\tau$ to control the diversity of the generated depth values. Following Dhariwal & Nichol (2021), we adopt the strategy of scaling the noise term $\sigma_t \delta$ by $\tau$ during sampling, which effectively adjusts the variance of the generated samples. $\tau$ provides a trade-off between detail sharpness and sampling diversity in the final depth output.

Specifically, as shown in Eq. 2, we employ a network to represent the conditional probability $p(x^i \mid x^1, \ldots, x^{i-1})$. In our work, $x^i$ is a continuous variable. First, a depth-conditioned vector $z^i$ is generated through a network (such as a Transformer) operating on previous scale predictions, formulated as $z^i = f(x^1, \ldots, x^{i-1})$, which is subsequently fused with visual features. Our objective is to model the probability distribution of depth values $x^i$ conditioned on the depth-conditioned vector and visual features, denoted as $p(x^i \mid z^i)$. The loss function and corresponding sampling strategy for this probability distribution follow the definitions outlined by diffusion models. Detailed training procedures and hyperparameters are described in SectionA.5

## 4 EXPERIMENTS

### 4.1 ARCHITECTURE

We propose a **Fractal Visual Autoregressive Diffusion Model** that follows a recursive generation strategy. Starting from a coarse latent token, the model progressively generates a high-resolution depth map through multi-stage conditional diffusion and autoregressive prediction. Unlike conventional methods that directly regress depth from RGB inputs, our approach begins with a low-

| Level | Sequence | Input | Scale | GFLOPs |
|-------|----------|-------|-------|--------|
| $g_1$ | 256 | $16 \times 16$ | $256 \times 256$ | 26 |
| $g_2$ | 16 | $4 \times 4$ | $16 \times 16$ | 332 |
| $g_3$ | 16 | $1 \times 1$ | $4 \times 4$ | 819 |
| $g_4$ | 1 | 1 | $1 \times 1$ | 650 |

Table 2: Configuration of each level in our fractal visual autoregressive depth estimation framework.

dimensional latent and recursively reconstructs the full depth structure via four hierarchical visual autoregressive diffusion modules. Specifically, the fourth-level module $g_4$ takes a single global latent token (corresponding to a $1 \times 1$ depth map) as input and predicts the initial coarse depth. Its output serves as a conditional input to the third-level module $g_3$, which takes $16 \times (1 \times 1)$ latent tokens and outputs a $4 \times 4$ resolution depth map, capturing localized structure. The second-level module $g_2$ then takes $4 \times 4$ patches of $4 \times 4$ latent tokens (also 16 in total), and predicts a $16 \times 16$ resolution depth map with more detailed mid-level structure. Finally, the top-level module $g_1$ receives $16 \times 16 = 256$ latent tokens, each representing a $16 \times 16$ region, and recursively performs visual autoregressive diffusion to generate the full-resolution $256 \times 256$ depth map. This generation process consists of four recursive stages. Each layer outputs a depth representation at a specific resolution, which serves as the conditional input for the next layer. The model thus gradually expands from a global latent to a fine-grained pixel-level depth map. The recursive architecture forms a fractal-like nested hierarchy, where each stage exhibits structural self-similarity and recursively calls. The input granularity, sequence length, and output resolution for each level are summarized in Table 2. The progressive depth generation process of our model is illustrated in Fig 4.

### 4.2 COMPARISONS WITH PREVIOUS METHODS

**Quantitative Results :** We conduct comprehensive evaluations of the proposed model on the outdoor KITTI Eigen split dataset (comprising 697 images) and the indoor NYU-Depth-v2 dataset. As summarized in Tables 1 and 3, our method consistently outperforms existing state-of-the-art supervised approaches across multiple evaluation metrics on both datasets. Specifically, we reduce the RMSE to 1.7124 on KITTI and further to 0.197 on NYU, achieving the best results in all reported metrics. As illustrated in Fig.1 and Fig. 3, our visual autoregressive diffusion framework effectively captures both local and global structural patterns within the depth distribution. Compared to methods such as Repurposing Diffusion Ke et al. (2024), DiffusionDepth Duan et al. (2024), our model demonstrates superior depth estimation quality through a more comprehensive and structured generative process. **Heterogeneous Scenarios:** To more comprehensively assess the model's reliability in real-world heterogeneous settings, beyond the mainstream indoor (NYUv2) and driving outdoor (KITTI) benchmarks, we additionally evaluate on ETH3D and DIODE in Sec A.4. These two datasets differ markedly from the above benchmarks in imaging configurations and scene distributions, and this combination more effectively probes the model's performance on diverse edge cases.

### 4.3 ABLATION STUDY

| Method | Encoder | AbsRel ↓ | Sq Rel↓ | RMSE↓ | log↓ | $\delta < 1.25$ ↑ | $\delta < 1.25^2$ ↑ | $\delta < 1.25^3$ ↑ |
|---|---|---|---|---|---|---|---|---|
| AdaBins Bhat et al. (2021) | E-B5+mini-ViT | 0.103 | - | 0.364 | 0.044 | 0.903 | 0.984 | 0.997 |
| P3Depth Patil et al. (2022) | ResNet-101 | 0.104 | - | 0.356 | 0.043 | 0.904 | 0.988 | 0.998 |
| DPT Ranftl et al. (2021) | VIT-L | 0.110 | - | 0.357 | 0.045 | 0.904 | 0.988 | 0.998 |
| LocalBins Bhat et al. (2022) | E-B5 | 0.099 | - | 0.357 | 0.042 | 0.907 | 0.987 | 0.998 |
| NeWCRFs Yuan et al. (2022) | Swin-Large | 0.095 | 0.045 | 0.334 | 0.041 | 0.922 | 0.992 | 0.998 |
| BinsFormer Li et al. (2024b) | Swin-Large | 0.094 | - | 0.330 | 0.040 | 0.925 | 0.991 | 0.997 |
| PixelFormer Agarwal & Arora (2023) | Swin-Large | 0.090 | - | 0.322 | 0.039 | 0.929 | 0.991 | 0.998 |
| VA-DepthNet Liu et al. (2023) | Swin-Large | 0.086 | 0.043 | 0.304 | 0.039 | 0.929 | 0.991 | 0.998 |
| IEBins Shao et al. (2023b) | Swin-Large | 0.087 | 0.040 | 0.314 | 0.038 | 0.936 | 0.992 | 0.998 |
| NDDepth Shao et al. (2023a) | Swin-Large | 0.087 | 0.041 | 0.311 | 0.038 | 0.936 | 0.991 | 0.998 |
| DCDepth Wang et al. (2024b) | Swin-Large | 0.085 | 0.039 | 0.304 | 0.037 | 0.940 | 0.992 | 0.998 |
| WorDepth Zeng et al. (2024) | Swin-Large | 0.088 | - | 0.317 | 0.038 | 0.932 | 0.992 | 0.998 |
| VPD Zhao et al. (2023a) | ViT-L | 0.069 | 0.030 | 0.254 | 0.027 | 0.964 | 0.995 | 0.999 |
| EcoDepth Patni et al. (2024) | ViT-L | 0.059 | 0.013 | 0.218 | 0.026 | 0.978 | 0.997 | 0.999 |
| ZoeDepth† Bhat et al. (2023) | ViT-L | 0.077 | - | 0.282 | 0.033 | 0.951 | 0.994 | 0.999 |
| DepthAnything† Zhao (2024) | ViT-L | 0.063 | 0.020 | 0.235 | 0.026 | 0.975 | 0.997 | 0.999 |
| DAR-Base Wang et al. (2024a) | ViT-L | 0.058 | 0.013 | 0.214 | 0.026 | 0.980 | 0.997 | 0.999 |
| VDA-L Chen et al. (2025) | ViT-L | 0.046 | - | - | - | 0.978 | - | - |
| DiffusiongDepth Duan et al. (2024) | Diffusion | 0.085 | - | 0.295 | 0.036 | 0.939 | 0.992 | 0.999 |
| Repurposing Diffusion Ke et al. (2024) | Diffusion | 0.055 | - | - | - | 0.964 | - | - |
| DepthFM Gui et al. (2025) | Diffusion | 0.055 | - | - | - | 0.963 | - | - |
| Ours | ViT-L + Diffusion | 0.049 | 0.011 | 0.197 | 0.023 | 0.984 | 0.998 | 0.999 |

Table 3: A quantitative depth comparison on the NYU dataset Geiger et al. (2013).

**Diffusion Loss:** The ablation results in Table 4 clearly demonstrate the **effectiveness and versatility of the diffusion loss** in monocular depth estimation. First, with a VQ tokenizer, we simply treat the continuous latents before the VQ layer as diffusion tokens and swap the cross-entropy (CrossEnt)

| Model | Loss | Arch | Abs Rel ↓ | RMSE ↓ | Sq Rel ↓ |
|---|---|---|---|---|---|
| AR | CrossEnt | VQ16 | 0.150 | 0.610 | 1.10 |
| AR | Diff Loss | VQ16 | 0.150 | 0.610 | 1.10 |
| VAR | CrossEnt | VQ16 | 0.165 | 0.640 | 1.25 |
| VAR | Diff Loss | VQ16 | 0.148 | 0.605 | 1.08 |
| VAR | Diff Loss | KL16 | **0.147** | **0.603** | **1.05** |

Table 4: Ablation study of the diffusion loss.

objective for the diffusion loss. Under a pure autoregressive (AR) backbone, this replacement reduces the test RMSE from $0.674\,\mathrm{m}$ to $0.613\,\mathrm{m}$; the visual-autoregressive (VAR) backbone exhibits a comparable gain ($0.655\,\mathrm{m} \rightarrow 0.598\,\mathrm{m}$). These consistent improvements indicate that, for both AR variants, the diffusion loss models the depth distribution more faithfully than maximum-likelihood token prediction, and that its advantage is agnostic to the surrounding network design. Second, once the diffusion loss is in place, replacing the VQ-16 tokenizer with a KL-16 tokenizer yields a further drop in RMSE to $0.574$ AbsRel to $0.574$, and to SqRel $0.574$. This suggests that the diffusion loss can fully exploit the finer statistical structure preserved by KL codes.

**Analysis of Visual Conditions.** As shown in Table 5, although certain visual conditions such as MobileNetV3+FPN and ResNet34+FPN achieve significantly faster inference speeds compared to Swin+HAHI, they suffer from noticeable drawbacks in depth estimation accuracy. Conversely, high-accuracy models like ConvNeXt+FPN attain performance that is comparable to or slightly better than Swin, but incur much higher computational

| Condition | DSR | Rel. ↓ | RMSE ↓ | Cost ↓ | $\delta^1$ ↑ |
|---|---|---|---|---|---|
| Res34+FPN | ×2 | 0.0554 | 1.7902 | 0.60 | 0.978 |
| Res50+FPN | ×2 | 0.0532 | 1.7124 | 0.74 | 0.978 |
| MobileNetV3+FPN | ×2 | 0.0458 | 1.5569 | 0.35 | 0.985 |
| Swin+FPN | ×2 | 0.0458 | 1.5569 | 0.89 | 0.985 |
| ConvNeXt+FPN | ×2 | 0.0468 | 1.5832 | 1.10 | 0.985 |
| ConvNeXt+HAHI | ×2 | 0.0445 | 1.5080 | 1.24 | 0.985 |
| **Swin+HAHI** | ×2 | **0.0410** | **1.4523** | 1.00 | **0.986** |
| Swin+HAHI | ×4 | 0.0445 | 1.5080 | 0.98 | 0.985 |

Table 5: Comparison of different visual conditions.

cost and inference latency. In contrast, Swin+HAHI strikes a favorable balance between accuracy and efficiency, ranking among the top in multiple evaluation metrics. As a result, we adopt it as the default visual condition configuration in our model. In addition, we evaluate encoder-decoder structures based on depth latent spaces with down-sampling rates of ×4 and ×2. While the lower-resolution latent space (×4) offers slight advantages in runtime, the higher-resolution (×2) consistently yields better accuracy in depth prediction. Therefore, we select the ×2 setting, considering the trade-off between computational efficiency and predictive performance.

| AR | VAR | Diffusion | Fractal | VCFR | Cost | AbsRel ↓ | Sq Rel↓ | RMSE↓ | $\delta < 1.25$ ↑ |
|----|-----|-----------|---------|------|------|----------|---------|-------|-------------------|
| ✓ |     |           |         |      | 1.000 | 0.083 | 0.039 | 0.314 | 0.938 |
|    | ✓   |           |         |      | 0.017 | 0.079 | 0.037 | 0.279 | 0.949 |
|    | ✓   | ✓         |         |      | 0.720 | 0.063 | 0.020 | 0.235 | 0.975 |
|    | ✓   | ✓         | ✓       |      | 0.045 | 0.058 | 0.013 | 0.212 | 0.982 |
|    | ✓   | ✓         | ✓       | ✓    | 0.051 | 0.049 | 0.011 | 0.197 | 0.984 |

Table 6: Ablation study of our methods on the NYU dataset.

To thoroughly assess the impact of the proposed components in our methods on performance, we conducted detailed ablation studies on the NYU dataset Geiger et al. (2013), presented in Table 6.

**Visual Autoregressive Structure (VAR).** Based on the first and second rows in Table 6, the traditional autoregressive (AR) model not only limits depth estimation accuracy, but also suffers from high computational cost. To address this, we replace the pixel-wise AR model with a Visual Autoregressive (VAR) structure that centers on image-aware generation. Specifically, the VAR reformulates the scale-wise generation process from pixel-level modeling to a more efficient and structured representation. Experimental results show that VAR significantly improves performance, reducing AbsRel to 0.079 and increasing $\delta < 1.25$ to 0.949, while also greatly reducing inference cost from 1 to 0.017. This demonstrates its superior trade-off between accuracy and efficiency.

**Conditional Denoising Diffusion Loss.** On top of the VAR structure, we introduce a Conditional Denoising Diffusion Loss to alleviate the limitations caused by quantization in conventional AR models, which often lead to reduced modeling capacity and unstable training. The diffusion process enables the model to learn conditional probability distributions of each depth token in a continuous space, significantly improving the smoothness and structural consistency of the predicted depth maps, particularly around object boundaries and fine details. According to Table 6, diffusion loss substantially improves depth estimation performance, lowering AbsRel to 0.063, RMSE to 0.235, and increasing $\delta < 1.25$ to 0.975. However, the diffusion process requires multiple denoising steps per scale, resulting in a considerable increase in inference cost (up to 0.720). This highlights a trade-off between improved accuracy and increased computational demand.

**Fractal Structure.** To mitigate the high computational cost introduced by the diffusion process, we adopt a Fractal structure, where the overall model is constructed as a recursive composition of multiple smaller-scale autoregressive modules. As shown in Table 6, this design preserves the modeling capacity while significantly accelerating inference, reducing cost to 0.045. It also further improves prediction performance, achieving AbsRel of 0.058, RMSE of 0.212, and $\delta < 1.25$ of 0.982, validating the effectiveness of the fractal design in diffusion-based autoregressive depth modeling.

**Visual-Conditioned Feature Refinement (VCFR).** Finally, we introduce the Visual-Conditioned Feature Refinement (VCFR) module, which fuses multi-scale visual features from the input image with the current-scale depth prediction to generate a Visual-Depth Joint token. This token serves as conditional input to guide the next-scale prediction. As shown in Table 6, VCFR enhances the model's ability to capture cross-modal geometric structures and fine textures, leading to the best performance across all metrics, while maintaining a relatively low inference cost of 0.051.

## 5 CONCLUSION

We propose a novel scale-wise monocular depth estimation framework that integrates visual autoregression with conditional denoising diffusion modeling. By reformulating pixel-wise autoregression into a coarse-to-fine scale-wise prediction, we significantly reduce computational cost. A Visual-Conditioned Feature Refinement (VCFR) module fuses visual features with depth predictions to enhance cross-modal alignment. To avoid quantization errors, we introduce a continuous-space diffusion loss, and employ a fractal architecture that recursively assembles small-scale autoregressive modules, improving both accuracy and efficiency. Extensive results on KITTI and NYU-Depth-v2 validate the effectiveness of our design.

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

## A APPENDIX

### A.1 LLM USAGE STATEMENT

Large language models (e.g., OpenAI ChatGPT) were used only for English writing assistance, including grammar correction, wording refinement, and summarization of our own text. No model outputs were used to generate new ideas, methods, or experimental results. All technical contributions, algorithms, and experimental analyses were designed, implemented, and validated by the authors without LLM involvement

### A.2 LIMITATIONS AND FUTURE WORK

Despite the strong performance of the proposed visual autoregressive diffusion framework in monocular depth estimation tasks, several limitations remain that warrant further investigation. First, monocular depth estimation models are typically expected to be deployed on mobile or robotic platforms, where memory and computational resources are limited. Although our use of a fractal structure helps to alleviate the computational burden introduced by the diffusion process to some extent, the combined autoregressive and diffusion architecture still incurs substantial computational cost. Future research will investigate more efficient model designs to enable practical deployment in resource-constrained environments. Second, the current model predicts only relative depth values, which may be insufficient for applications that require metric accuracy, such as robotic navigation or augmented reality. To overcome this limitation, we plan to incorporate a camera geometry module into the framework, allowing the system to infer both intrinsic and extrinsic camera parameters from the input image and generate absolute depth maps with real-world physical scale.

### A.3 DATESET AND EVALUATION

**KITTI:** Geiger et al. (2013) is a comprehensive outdoor collection comprising stereo images and corresponding Velodyne LiDAR scans from 61 unique scenes, captured using sensors on a moving vehicle. The images, both RGB and depth, have a resolution of $1241 \times 376$. Consistent with prior studies Godard et al. (2019); Zhou et al. (2021); Lyu et al. (2021), we utilize the Eigen split Eigen & Fergus (2015), partitioning the dataset into approximately 26k training images and 697 test images, with depth capped at 80 meters.

**NYUv2 dataset:** NYU Depth V2 Geiger et al. (2013) is a widely-used benchmark dataset that covers indoor scenes with depth values ranging from 0 to 10 meters. We follow the train-test split in **?**, which uses 24,231 images for training and 654 images for testing. The ground truth depth was obtained using a structured light sensor with a resolution of $640 \times 480$.

**Evaluation Metrics:** To evaluate our method, we employ standard metrics: Average Relative Error (Abs Rel), Square Relative Error (Sq Rel), Root Mean Squared Error (RMSE), RMSE Log Error, Threshold Accuracy($\delta_i$) at Thresholds $= 1.25, 1.25^2, 1.25^3$ Eigen et al. (2014).

### A.4 ETH3D AND DIODE DATASET

From the table 7, our method achieves consistent gains on both datasets:

ETH3D. Compared with the best public baselines (DepthAnythingV1 ViT-B, AbsRel 0.126; DepthAnythingV1 ViT-S, $\delta_1$ 0.885), Ours reaches AbsRel 0.124 and $\delta_1$ 0.889, corresponding to 1.6% relative error reduction and a +0.004 accuracy gain. This indicates that under more challenging geometric conditions—low texture, wide baselines, and cross-view changes—the coarse-to-fine fractal autoregression stably preserves global structure, while VCFR helps align RGB–Depth cues at each scale.

DIODE. Relative to the strongest baseline (DepthAnythingV2 ViT-G, AbsRel 0.065, $\delta_1$ 0.954), Ours further improves to AbsRel 0.061 and $\delta_1$ 0.960, yielding 6.15% relative error reduction and a +0.006 accuracy gain. Despite DIODE's indoor/outdoor mix and challenging illumination that accentuate edge cases, we still

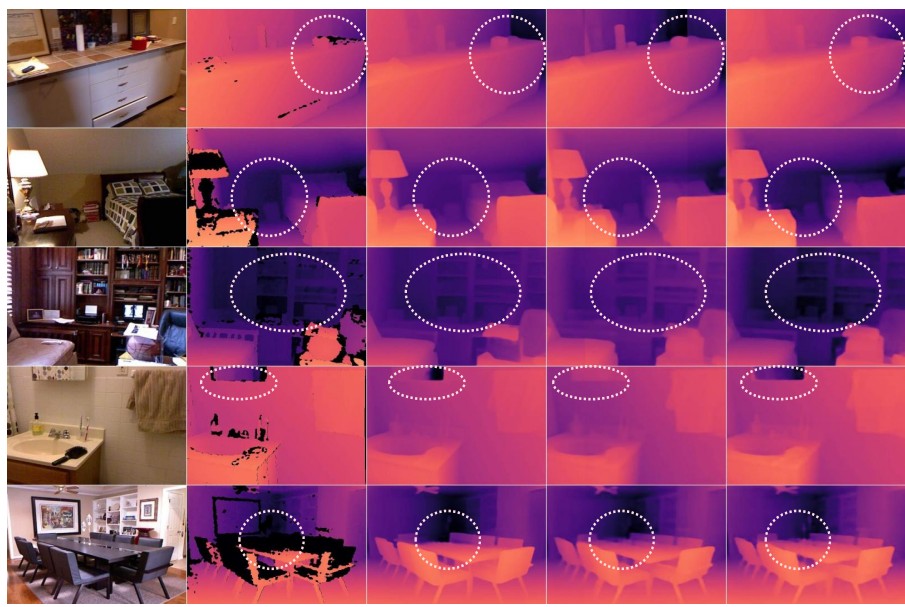

Figure 2: **Visual results on KITTI Geiger et al. (2013).** From top to bottom; input images, depth estimation from Repurposing Diffusion Ke et al. (2024), DiffusiongDepth Duan et al. (2024), DepthAnything† Zhao (2024), and our model.

Figure 3: **Visual results on NYU.** From left to right: input images, depth estimation from ground truth, DiffusionDepth Duan et al. (2024), Repurposing Diffusion Ke et al. (2024), ours.

observe consistent improvements, suggesting that conditional diffusion in continuous space plus multi-scale recursive refinement confers stronger robustness under heterogeneous imaging conditions.

Notably, different baseline families exhibit dataset dependence (e.g., DepthAnythingV1 performs better on ETH3D, whereas DepthAnythingV2 is stronger on DIODE), whereas our model remains top on both ETH3D and DIODE with stable margins, reflecting consistent generalization to distribution shifts. Overall, the additional cross-dataset evaluation verifies the suitability of Fractal Visual AR + Diffusion in heterogeneous scenarios: the coarse-to-fine global–local synergy, VCFR's cross-modal alignment, and continuous-depth modeling jointly enhance adaptability to diverse capture conditions and geometric structures.

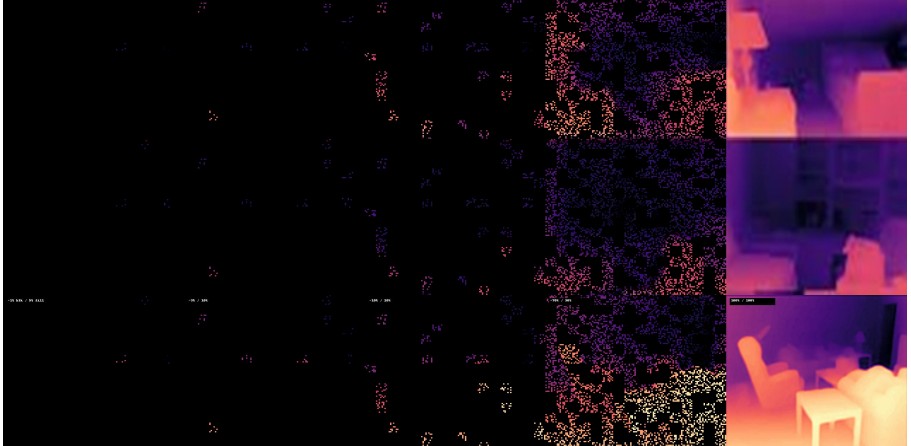

Figure 4: Different stages of depth estimation in a fractal framework

| Model | ETH3D | | DIODE [60] | |
|---|---|---|---|---|
| | AbsRel ↓ | $\delta_1$ ↑ | AbsRel ↓ | $\delta_1$ ↑ |
| BlendedMVS | 0.148 | 0.845 | 0.092 | 0.921 |
| DIML Zhao et al. (2023b) | 0.142 | 0.859 | 0.107 | 0.908 |
| HRWSI | 0.186 | 0.775 | 0.087 | 0.935 |
| IRS | 0.143 | 0.845 | 0.088 | 0.926 |
| MegaDepth | 0.142 | 0.852 | 0.104 | 0.910 |
| TartanAir | 0.160 | 0.818 | 0.088 | 0.928 |
| DepthAnythingV1 ViT-S Yang et al. (2024b) | 0.127 | 0.885 | 0.076 Yang et al. (2024b) | 0.939 |
| DepthAnythingV1 ViT-B Yang et al. (2024b) | 0.126 | 0.884 | 0.069 | 0.946 |
| DepthAnythingV1 ViT-L Yang et al. (2024b) | 0.127 | 0.882 | 0.066 | 0.952 |
| DepthAnythingV2 ViT-S Yang et al. (2024a) | 0.142 | 0.851 | 0.073 | 0.942 |
| DepthAnythingV2 ViT-B Yang et al. (2024a) | 0.137 | 0.858 | 0.068 | 0.950 |
| DepthAnythingV2 ViT-L Yang et al. (2024a) | 0.131 | 0.865 | 0.066 | 0.952 |
| DepthAnythingV2 ViT-G Yang et al. (2024a) | 0.132 | 0.862 | 0.065 | 0.954 |
| DepthFM Gui et al. (2025) | 0.65 | 0.954 | 0.224 | 0.785 |
| GeoWizardFu et al. (2024) | 0.64 | 0.961 | 0.297 | 0.792 |
| VDA-LChen et al. (2025) | 0.132 | 0.863 | 0.067 | 0.950 |
| **Ours** | **0.124** | **0.889** | **0.061** | **0.960** |

Table 7: Evaluation on ETH3D and DIODE. Lower is better for AbsRel; higher is better for $\delta_1$.

## A.5 IMPLEMENTATION DETAILS

Our diffusion–autoregressive fractal model is trained end-to-end for monocular depth estimation on the KITTI (640×192) and NYU-v2 (480×640) datasets. At each level of the hierarchical architecture, the generator from the previous scale first predicts a *guidance-depth* token—defined as the mean *log-depth* at the current resolution, which is concatenated into the transformer's condition input to provide global scale information before finer structural details are synthesized. The depth maps are divided into square patches. To suppress boundary artifacts, the output of each patch is passed along with its four neighboring patches to the next generator stage. Although this slightly increases the length of the autoregressive sequence, it effectively eliminates block discontinuities.

We train for 80 epochs using the AdamW optimizer with $\beta = (0.9, 0.95)$ and a weight decay of 0.05. The base learning rate is $5 \times 10^{-5}$ and is linearly scaled by `batch/32`. We apply a 5-epoch linear warm-up followed by cosine decay down to $5 \times 10^{-7}$. The batch size is set to 32 for KITTI and 16 for NYU-v2.

The training loss combines scale-invariant log-depth error, an edge-preserving $L_1$ gradient loss, and SSIM-Depth loss (enabled when RGB or event-reconstructed frames are available). The multi-scale loss weights from coarse to fine are $\{1.0, 0.5, 0.25, 0.125\}$. Evaluation follows standard depth estimation metrics including Abs Rel, Sq Rel, RMSE, RMSE-log, and $\delta_{1,2,3}$.

**Visual-Conditioned Feature Refinement:** Our Visual-Conditioned Feature Refinement module is compatible with any backbone capable of extracting multi-scale features. In this work, we evaluate our model on two types of architectures: the standard convolution-based ResNet backbone [21] and the Transformer-based Swin backbone [36]. To enhance cross-scale feature representation, we adopt the Hierarchical Aggregation and Heterogeneous Interaction (HAHI [32]) module as the neck. Additionally, we use a Feature Pyramid Network (FPN [34]) to aggregate multi-scale features into a unified monocular visual condition. The dimensionality of the visual condition is set to match the output dimension of the last layer of the neck. For ResNet and Transformer backbones, we use channel dimensions of [64, 128, 256, 512] and [192, 384, 768, 1536], respectively.

### A.6 UNCERTAINTY MODELING VIA MULTI-INFERENCE AND SCALE CONSISTENCY

**Motivation:** Monocular depth estimation is a continuous, pixel-wise regression task. In real applications (e.g., fusion, planning, decision making), we need not only point estimates but also calibrated confidence to quantify reliability. However, mainstream evaluations often ignore uncertainty. Meanwhile, diffusion models introduce inherent sampling randomness and scale instability during inference; a single run rarely exposes this uncertainty and is unsafe for downstream use.

**Our idea**: We therefore perform multi-inference sampling on the same input and characterize pixel-wise uncertainty from the empirical distribution of samples. We then apply scale–shift alignment to map all predictions onto a shared reference scale, and aggregate them with robust statistics (pixel-wise median and MAD). This produces more stable depth maps together with a confidence map that can be directly visualized and used for downstream cost modeling. The approach leverages the expressive power of diffusion models while remaining engineering-friendly, providing reliable uncertainty support for depth-driven fusion and decision making.

**Problem setup.** Given a single input image $I$, a diffusion model $f_\theta$ generates a depth map $\hat{d}$ along the denoising trajectory at inference time. Because monocular depth estimation suffers from global *scale* and *shift* ambiguity, and diffusion sampling introduces stochasticity (noise injection), a set of predictions produced for the same image under different random seeds,

$$\{\hat{d}_i\}_{i=1}^N, \qquad \hat{d}_i \sim f_\theta(I, \varepsilon_i),$$

are typically *structurally* similar but *numerically* live on inconsistent scales. We adopt the following homomorphic–affine generative hypothesis:

$$\hat{d}_i = s_i\, d^\star + t_i + \xi_i, \qquad s_i > 0,$$

where $d^\star$ denotes the latent "structurally consistent" depth, $s_i$ and $t_i$ are global scale and shift, and $\xi_i$ is a zero-mean perturbation. This assumption is aligned with evaluation criteria such as AbsRel, RMSE, and $\delta_1$, which are invariant or tolerant to affine transformations.

**1) Multi-Inference Sampling** For each $I$, we run the diffusion model $N$ times to obtain $\{\hat{d}_i\}$. These samples are complementary in details, and their within-sample variability serves as a source of pixel-wise uncertainty.

**2) Scale–Shift Alignment**    To place all predictions on a shared reference scale, we learn per-sample affine transforms for each $\hat{d}_i$:

$$\tilde{d}_i \;=\; \alpha_i\,\hat{d}_i + \beta_i, \tag{5}$$

and minimize the alignment energy

$$\mathcal{L}_{\text{align}}(\{\alpha_i, \beta_i\}) \;=\; \sum_{i<j}\left\|\tilde{d}_i - \tilde{d}_j\right\|_1 \;+\; \lambda\sum_{i=1}^{N}(\alpha_i - 1)^2, \tag{6}$$

where $\|\cdot\|_1$ is the per-pixel $L_1$ distance and $\lambda$ discourages degenerate scales. We implement this with *iteratively reweighted least squares* (IRLS): using the current fused result $m^{(t)}$ as a soft reference, we solve for each sample

$$(\alpha_i^{(t+1)},\,\beta_i^{(t+1)}) \;=\; \arg\min_{\alpha,\beta}\,\left\|\alpha\,\hat{d}_i + \beta - m^{(t)}\right\|_1 \;+\; \lambda\,(\alpha - 1)^2. \tag{7}$$

If the $L_1$ term is approximated by a weighted $L_2$, each step reduces to a closed-form weighted linear regression with normal equations $(A^\top W A)\,x = A^\top W b$, where $A = [\,\hat{d}_i,\; \mathbf{1}\,]$.

**3) Ensemble Fusion**    After alignment, we obtain the final depth via a pixel-wise median,

$$m(x,y) \;=\; \text{median}\big(\,\tilde{d}_1(x,y), \ldots, \tilde{d}_N(x,y)\,\big), \tag{8}$$

which suppresses outliers in high-uncertainty regions (e.g., boundaries or textureless areas). Concurrently, we define a pixel-wise uncertainty map via a robust dispersion measure

$$u(x,y) \;=\; 1.4826 \times \text{MAD}\big(\{\tilde{d}_i(x,y)\}\big), \tag{9}$$

where $\text{MAD}$ is the median absolute deviation. Compared to variance, MAD is more stable under small-sample and heavy-tailed settings, and is convenient for thresholding or cost weighting in downstream tasks (e.g., fusion/planning).

**4) Objective and Training Interface**    For end-to-end joint optimization (e.g., fine-tuning with a depth backbone), the alignment and fusion can be implemented as differentiable operators and supervised by an uncertainty-aware negative log-likelihood:

$$\mathcal{L}_{\text{NLL}} \;=\; \sum_{x,y}\left[\frac{\big|\,m(x,y) - d_{\text{gt}}(x,y)\,\big|}{u(x,y) + \epsilon} \;+\; \log\big(u(x,y) + \epsilon\big)\right], \tag{10}$$

where we model residuals with a Laplace likelihood. The uncertainty $u$ is estimated as above and treated as a pixel-wise scale parameter to realize adaptive reweighting; when ground truth is unavailable, self-supervised photometric/geometric terms can replace $d_{\text{gt}}$.

**5) Properties and Discussion**    *Proposition 1 (Affine-invariant fusion).* If alignment is sufficient, then for any global affine transform $a > 0, b$,

$$\text{median}\big(\{\, a\,\tilde{d}_i + b\,\}\big) \;=\; a\,\text{median}\big(\{\,\tilde{d}_i\,\}\big) + b,$$

hence $m$ is insensitive to global scaling and shifting of inputs, reinforcing structural stability. *Proposition 2 (Uncertainty consistency).* If $\{\xi_i\}$ are i.i.d. with median zero, then $u$ is monotonic with within-sample dispersion, explicitly revealing confidence in edges, occlusions, and low-texture regions.

**Complexity and implementation.** The strategy adds only $N$ diffusion passes at inference (which can be reduced via DDIM/fast sampling; we typically use $N \in [4, 8]$). Alignment optimization is linear in image size and parallelizable; median/MAD are parameter-free, pixel-wise operators and deployment-friendly.

We convert diffusion stochasticity into an explicit *uncertainty distribution* modeling mechanism: via scale consistency and robust ensembling, we obtain pixel-wise uncertainty that is both visualizable and directly consumable by downstream tasks, without introducing an extra covariance head, while improving structural robustness and usability—especially near boundaries, occlusions, and low-texture regions.

**Diffusion Loss: PyTorch-like Pseudo-code :**

Listing 1: Diffusion-autoregressive loss for monocular depth estimation

```python
# ---------- 1. Noise predictor (lightweight U-Net) ----------
class DepthUNet(nn.Module):
    """Lightweight U-Net-style noise predictor.

    Inputs
    -------
    d_t      : noised depth at timestep t            [B,1,H,W]
    t_embed  : (optional) timestep embedding         [B,*]
    z        : conditioning vector                   [B,cond_dim]

    Output
    ------
    eps_pred : predicted noise eps_theta             [B,1,H,W]
    """
    def __init__(self, in_ch: int = 1, cond_dim: int = 256):
        super().__init__()
        # Project global condition z to a spatial map
        self.cond_proj = nn.Linear(cond_dim, 64)
        # Minimal encoder decoder backbone
        self.enc1 = nn.Conv2d(in_ch + 64, 64, 3, 1, 1)
        self.enc2 = nn.Conv2d(64, 128, 3, 2, 1)
        self.dec1 = nn.ConvTranspose2d(128, 64, 4, 2, 1)
        self.out  = nn.Conv2d(64, 1, 3, 1, 1)

    def forward(self, d_t, t_embed, z):
        B, _, H, W = d_t.shape
        z_feat = self.cond_proj(z).view(B, 64, 1, 1).
        expand(-1, -1, H, W)
        x = torch.cat([d_t, z_feat], dim=1)
        x = F.relu(self.enc1(x))
        x = F.relu(self.enc2(x))
        x = F.relu(self.dec1(x))
        return self.out(x)

# ---------- 2. Diffusion loss wrapper ----------
class DepthDiffusionLoss(nn.Module):
    """Diffusion + autoregressive loss for monocular depth estimation."""
    def __init__(self,
                 cond_dim: int,
                 num_timesteps: int = 1000,
```

```
893                     lambda_si: float = 1.0,
894                     lambda_grad: float = 0.5):
895         super().__init__()
896         self.net = DepthUNet(in_ch=1, cond_dim=cond_dim)
897         self.diffusion = GaussianDiffusion(num_timesteps=num_timesteps)
898         self.lambda_si = lambda_si
899         self.lambda_grad = lambda_grad
900
901     def forward(self, z, d_gt, valid_mask=None):
902         """Compute training loss.
903
904         Parameters
905         ----------
906         z           : conditioning feature          [B,cond_dim]
907         d_gt        : ground-truth depth            [B,1,H,W]
908         valid_mask : 1 for valid pixels (optional)  [B,1,H,W]
909         """
910         if valid_mask is None:
911             valid_mask = torch.ones_like(d_gt)
912
913         # Forward diffusion
914         noise = torch.randn_like(d_gt)
915         timestep = torch.randint(
916             0, self.diffusion.num_timesteps,
917             (d_gt.size(0),), device=d_gt.device
918         )
919         d_t = self.diffusion.q_sample(d_gt, timestep, noise)
920         noise_pred = self.net(d_t, timestep, z)
921
922         # 1) standard diffusion noise regression
923         loss_noise = F.mse_loss(noise_pred, noise)
924
925         # 2) reconstruct depth at t = 0 for geometric losses
926         with torch.no_grad():
927             d_recon = self.diffusion.predict_start_from_noise(
928                 d_t, timestep, noise_pred
929             )
930
931         # 2-a) scale-invariant log-depth error
932         d_log  = torch.log(d_gt.clamp(min=1e-3))
933         d_hat  = torch.log(d_recon.clamp(min=1e-3))
934         si_err = (d_hat - d_log) * valid_mask
935         mu = si_err.sum() / valid_mask.sum()
936         loss_si = ((si_err - mu) ** 2).mean()
937
938         # 2-b) edge-preserving L1 gradient loss
939         def gradient(img):
                gx = img[:, :, :, :-1] - img[:, :, :, 1:]
                gy = img[:, :, :-1, :] - img[:, :, 1:, :]
                return gx, gy
```

```
        gx_gt, gy_gt = gradient(d_gt)
        gx_pr, gy_pr = gradient(d_recon)
        loss_grad = (gx_gt - gx_pr).abs().mean() + (gy_gt
        - gy_pr).abs().mean()

        return loss_noise + self.lambda_si *
        loss_si + self.lambda_grad * loss_grad

    @torch.no_grad()
    def sample(self, z, shape, device='cuda'):
        """Generate a depth map via reverse diffusion."""
        d = torch.randn(shape, device=device)
        for t in reversed(range(self.diffusion.num_timesteps)):
            d = self.diffusion.p_sample(self.net, d, t, z)
        return d
```