# OpenReview forum: "Recursive Autoregressive Depth Estimation with Continuous Token Modeling"
_ICLR.cc/2026/Conference — ICLR 2026 Conference Withdrawn Submission_

### Official Review · Reviewer_Gk6o · 2025-10-26

**Soundness:** 3
**Presentation:** 2
**Contribution:** 2
**Rating:** 4
**Confidence:** 5

**Summary:**

This paper proposes a new Fractal Visual AR + Diffusion framework for monocular depth estimation that combines autoregressive modeling with diffusion processes in a recursive, multi-scale architecture. Although effective, this article still has some deficiencies in terms of innovation, motivation and deployment performance discussion (e.g, speed and parameters), and has not yet reached the acceptance bar of ICLR.

**Strengths:**

1. Experimental results demonstrate superior results across multiple metrics on standard benchmarks.

2. Avoids quantization errors by working directly in continuous space, improving depth smoothness and accuracy.

3. The VCFR module effectively addresses the RGB-Depth modality mismatch problem.

**Weaknesses:**

1. The writing is not good, making the paper difficult to read. The figure of the fractal framework is too abstract.

2. The method seems like a combination of existing methods, making the paper less novel.

3. Authors claim that “To reclaim speed, we abstract the Visual AR unit into a reusable base generator and invoke it recursively, forming a self-similar fractal architecture that preserves modeling power while cutting the inference path.” However, despite improvements, the combined autoregressive-diffusion architecture still incurs substantial computational cost since the AR module needs to run multiple times. And the paper does not even show any performance or comparison on model speed.

**Questions:**

1. Motivation is not obvious. Using a continuous VAE is an easier way to model depth in a continuous latent space. Why use diffusion?

2. How is the zero-shot generalization performance of the proposed model? It seems that there are no experimental results.

---

### Official Review · Reviewer_ioyU · 2025-10-29

**Soundness:** 4
**Presentation:** 2
**Contribution:** 3
**Rating:** 4
**Confidence:** 2

**Summary:**

The paper proposes fractal visual AR combined with diffusion for monocular depth estimation. The idea is to replace pixel-wise AR with next-scale prediction (VAR), so that each token represents an entire feature map at a higher resolution. Additionally, a Visual-Conditioned Feature Refinement (VCFR) module is proposed, which fuses multi-scale RGB features with the current depth to form a 'Visual-Depth Joint' token. Depth is modelled continuously via conditional diffusion loss at every scale (rather than discrete codes). The entire approach is organised recursively (a 'fractal' stack g4→g1 from 1×1 to full resolution), with ablations demonstrating accuracy and efficiency gains. Experiments on KITTI, NYUv2, ETH3D, and DIODE demonstrate strong performance.

**Strengths:**

VAR avoids breaking spatial structure and fits dense prediction better than pixel AR. The combination with a fractal generative model is novel and intuitive. Overall, the paper make multiple incremental but solid contributions.

The fractal recursion (g4 → g1) and the VCFR pipeline are clearly described, with helpful figures and concrete-level configuration tables. The method is generally well explained and easy to follow. The same applies to the rest of the paper.

Evaluations on the KITTI, NYU, ETH3D and ScanNet datasets indicate that the approach achieves excellent performance. Tables 1 and 3 report clearly state-of-the-art results.

Ablation studies: Table 6 shows improvements to VAR, diffusion, fractal recursion and VCFR. The paper's contribution is well empirically justified.

**Weaknesses:**

The compute and latency are high. The level-wise GFLOPs are large (e.g. g3 = 819, g4 = 650) and diffusion adds many steps per scale. While the fractal design reduces cost relative to 'plain' diffusion, the absolute footprint appears heavy. The 'cost' column in Table 6 is unitless and difficult to relate to latency. Reporting wall clock time on KITTI/NYUv2 resolutions and hardware details would make sense. Methods such as [1] speed up conditional diffusion in a single step.

Training both an autoregressive and a diffusion model simultaneously is computationally intensive [2, 3]. The paper explains that combining them with VCFR reduces this complexity; however, the reason for this is unclear. More argumentation and explanation is necessary.

The writing in the paper is a minor issue. There are missing definitions and typos in the text and tables. For example, the citation for NYUv2 in Table 4 is incorrect; it should be Silberman et al. There are also inconsistencies in Table 4 with the text. Fractal recursion training is not defined. In the supplementary material, the formatting of Table 7 contains a citation.

[1] Garcia, G. M., Abou Zeid, K., Schmidt, C., De Geus, D., Hermans, A., & Leibe, B. (2025, February). Fine-tuning image-conditional diffusion models is easier than you think. In *2025 IEEE/CVF Winter Conference on Applications of Computer Vision (WACV)* (pp. 753-762). IEEE.

[2] Wang, J., Liu, J., Tang, D., Wang, W., Li, W., Chen, D., ... & Wu, J. (2025). Scalable autoregressive monocular depth estimation. In *Proceedings of the Computer Vision and Pattern Recognition Conference* (pp. 6262-6272).

[3] Saxena, S., Kar, A., Norouzi, M., & Fleet, D. J. (2023). Monocular depth estimation using diffusion models. *arXiv preprint arXiv:2302.14816*.

**Questions:**

The authors trained a VAR model with a diffusion loss. This is not a well-known concept. Could the authors please explain this in more detail?


What is the conceptual difference between your 'continuous token diffusion' approach and simply applying a Gaussian-likelihood regression loss to continuous depth values? Why is full diffusion modelling necessary?

---

### Official Review · Reviewer_Uznf · 2025-11-01

**Soundness:** 1
**Presentation:** 1
**Contribution:** 2
**Rating:** 2
**Confidence:** 3

**Summary:**

This paper presents a 'fractal' autoregressive diffusion method for monocular depth estimation. The paper identifies two limitations of autoregressive depth estimation models, which it aims to tackle: (1) autoregressive methods are inefficient and slow, and (2) there is a mismatch between the input and the output, as the model should map images to per-pixel depth outputs. To tackle the efficiency problem, the paper presents a fractal, 'next-scale prediction' framework where the model first predicts a depth output at a lower resolution and then gradually predicts higher-resolution outputs, decomposing the task into several smaller subproblems. To solve the modality mismatch, the paper proposes to fuse image features with depth predictions from the previous lower-resolution module and use this as conditional input to predict depth at a higher resolution, to provide the model with depth information as input to predict higher-resolution depth outputs. Experiments show that the overall model performs on par with existing state-of-the-art methods for monocular depth estimation.

**Strengths:**

1. The proposed method performs on par with existing state-of-the-art methods for monocular depth estimation (see Tab. 1 & Tab. 3). This demonstrates the effectiveness of the model’s design.
2. The paper properly evaluates the effectiveness of the individual components with an ablation study in Tab. 6. For instance, this table shows that the proposed VCFR module, which fuses image features with depth information to condition the next depth prediction, improves depth estimation performance across all metrics.
3. The idea to decompose autoregressive depth estimation into a set of smaller subproblems to improve efficiency is original and well-motivated. These smaller subproblems are more efficient to solve, potentially allowing for a more efficient overall method.

**Weaknesses:**

1.	The authors have changed the paper formatting with respect to the original ICLR template. (1) The page margins on the left and right side are smaller (making the text wider), (2) the vertical whitespace before and after some headers (e.g., headers for Sec. 2 an Sec 2.1 on L072 and L073, header for Sec. 3.3 in L208) has been reduced significantly, and (3) the whitespace around some equations has been reduced significantly (e.g., making the math from Eq. 1 actually overlap slightly with the text above and below the equation). This allows the authors to put more text in the 9 pages than other papers, which is an unfair advantage. Changing the margins also negatively impacts the presentation and readability of the paper.

2.	The abstract, in L015 states that the introduced framework predicts depth both accurately and efficiently, and L018 states that the presented approach greatly accelerates inference. Also in the introduction (L042-L046), the paper states that improving low inference speed is a key objective of the paper, and that the design significantly accelerates inference (L062). However, the efficiency of the overall proposed method is not properly evaluated nor compared to existing work. Tab. 6 only presents values for ‘computational cost’, but it is not clear what this cost actually represents (runtime, FLOPs, etc.) and what the absolute values are. Moreover, the paper does not compare the model’s efficiency to that of existing work.. As a result, the paper does not properly justify the efficiency claims. To verify these claims, the efficiency (runtime, FLOPs, num params, memory requirements) of the method should be experimentally evaluated and compared to the efficiency of existing state-of-the-art depth estimation methods. If the claims cannot be justified, the value of the paper decreases significantly.

3.	It is not clear what the motivation is for using autoregressive models for depth estimation. The introduction (L037) merely mentions that ‘autoregressive (AR) models have demonstrated impressive performance in image generation tasks’, but the authors do not mention why they believe autoregressive models would be suitable for depth estimation. The lack of a further elaboration of the motivation or rationale behind the approach limits the insights that can be gained from this paper.

4.	The operation of the overall method is not clearly explained. Sec. 3 only describes the high-level concepts underlying the method - e.g., factorization of the depth prediction (Eq. 1), progressively predicting the next scale (Eq. 2) - without concretely relating this to the different model components as visualized in Fig. 1. Sec. 4 briefly mentions that the model uses modules $\{g_4, g_3, g_2, g_1\}$ to use these concepts in practice, and it mentions the resolutions that they predict, but these modules $g_i$ are not referred to in Sec. 3. As such, it is not clear how these modules $g_i$ relate to the concepts that are explained in Sec. 3, and what exact operations the modules $g_i$ use. This makes it unclear how the method works, and harms reproducibility. The paper would be much clearer if Sec. 3 already clearly explained how the introduced concepts relate to the concrete model components that are used to implement these concepts, such that it is clear how the entire model operates, from the input image to the predicted depth output.

5.	In Figure 1, it is not clear what is can be seen in the ‘fractal framework’ visualization at the top-left of the figure, and how this ‘framework’ is related to the other aspects of the figure. There are no explanations of the colors, and no text labels for the inputs, outputs or individual blocks. This further harms understandability of the method and the paper as a whole.

There are some other minor weaknesses, which don’t significantly impact my rating:
* The paper uses `\cite{}` instead of `\citep{}` for citations (e.g., see L032-L033), which makes the citations appears as if they should be part of the main text. This limits readability. In the future, please use `\citep{}` where the authors’ names should not be part of the main text.
* In Tab. 1, ZoeDepth and DepthAnything have  $\dagger$ symbol next to their name, but the paper does not explain what this dagger symbol represents.
* In the caption of Tab. 3 (L345) the citation does not refer to the NYU dataset but to the KITTI dataset.
* L320 refers to Fig. 1 for qualitative results but this should be Fig. 2, which is in the appendix.

**Questions:**

While the paper presents a well-motivated idea and the method performs on par with existing state-of-the-art methods, the paper also has considerable weaknesses, which cause me to give a ‘reject’ rating. Most importantly, the authors have adjusted the formatting, the paper does not properly justify its efficiency claims, and the method is not explained sufficiently clearly. I would recommend the authors to address these issues, and also take into account the other comments and suggestions that I placed in the ‘weaknesses’ section.

---

### Official Review · Reviewer_BmzU · 2025-11-01

**Soundness:** 2
**Presentation:** 3
**Contribution:** 4
**Rating:** 6
**Confidence:** 4

**Summary:**

This paper considers a task of single view depth estimation. A new autoregressive framework with diffusion loss on continuous tokens is proposed. SOTA results are demonstrated on several benchmarks.

**Strengths:**

1) Novelty.
- Nobody done autoregressive models for depth estimation on continous tokens before (to the best of my knowledge)
- Interesint approach for fusion depth information and image information through VCFR module, which integrates infered depth tokens with image information

2) SOTA results
- The proposed model beats top models like DepthAnything v2 on several benchmarks

**Weaknesses:**

1) Missing details in description of the proposed method
a) Authors wrote that continous token lead to removal of quantization errors, but there is no information regarding reconstruction errors for vq-vae and kl-vae for depth. It should be demonstrated that discrete vae is worse for depth reconstruction compared to continous vae
b) Authors use real image datasets for model training. But such datasets contains missing depth pixels. When GT depth is encoded, such pixels influence the features. There is no information how such pixels are processed. This is the reason why many recent models lkie Marigold, Geowizard use synthetic data for training, where is no such problem.

2) SOTA results raises doubts due to limited training datasets.
Current state-of-the-art depth estimation methods either uses generative pre-trained models (trained on millions or billions of images) or uses very larger and diverse datasets, like DepthAnythingV2. The proposed model trained on KITTY and NYU-v2 datasets, which are rather small, and generalize excellently to ETH3D and DIODE. Thus it should be true breakthrough.

**Questions:**

Please, address the weaknesses I've mentioned in the respective section.

---

### Note · Authors · 2025-11-21

I have read and agree with the venue's withdrawal policy on behalf of myself and my co-authors.